

# 1 Black carbon (BC) in North Tibetan Mountain; Effect of

# 2 Kuwait fires on glacier

Jiamao Zhou[1,6], Xuexi Tie[1,2], Baiqing Xu[3], Shuyu Zhao[1], Mo Wang[3], Guohui Li[1],
Song Yang[3], Luyu Chang[4,5], Junji Cao[1]
[1]KLACP, SKLLQG, Institute of Earth Environment, Chinese Academy of Sciences, Xi'an 710061,
China
[2]Center for Excellence in Urban Atmospheric Environment, Institute of Urban Environment, Chinese
Academy of Sciences, Xiamen 361021, China
[3]Key Laboratory of Tibetan Environment Changes and Land Surface Processes, Institute ofTibetan
Plateau Research, Chinese Academy of Sciences, Beijing 100101, China
[4]Shanghai Meteorological Service, Shanghai, 200030, China
[5]Shanghai Key Laboratory of Meteorology and Health, Shanghai, 200030, China
[6] University of Chinese Academy of Sciences, Beijing 100049, China
*Correspondence to:* Xue Xi Tie (tiexx@ieecas.cn) or Baiqing Xu (baiqing@itpcas.ac.cn)



**Abstract.** The BC deposition on the ice core at Muztagh Ata Mountain, Northern Tibetan Plateau was
analyzed. Two sets of measurements were used in this study, which included the air samplings of BC
particles during 2004-2006 and the ice core drillings of BC deposition during 1986-1994. Two
numerical models were used to analyze the measured data. A global chemical transportation model
(MOZART-4) was used to analyze the BC transport from the source regions, and a radiative transfer
model (SNICAR) was used to study the effect of BC on snow albedo. The results show that during
1991-1992, there was a strong spike of the BC deposition at Muztagh Ata, suggesting that there was an
unusual emission in the upward region during this period. This high peak of BC deposition was
investigated by using the global chemical transportation model (MOZART-4). The analysis indicated
that the emissions from large Kuwait fires at the end of the first Gulf War in 1991 caused this high peak
of the BC concentrations and deposition (about 3-4 times higher than other years) at the Muztagh Ata
Mountain, suggesting that the upward BC emissions had important impacts on this remote site located
in Northern Tibetan Plateau. Thus, there is a need to quantitatively estimate the effect of surrounding
emissions on the BC concentrations in the northern Tibetan Plateau. In this study, a sensitive study with
4 individual BC emission regions (Central Asia, Europe, Persian Gulf, and South Asia) was conducted
by using the MOZART-4 model. The result suggests that during the "normal period" (non Kuwait
Fires), the largest effect was due to the Central Asia source (44%) during Indian monsoon period, while
during non-monsoon period, the largest effect was due to the South Asia source (34%). The increase of
radiative forcing increase (RFI) due to the deposition of BC on snow was estimated by using the
radiative transfer model (SNICAR). The results shows that under the fresh snow assumption, the
estimated increase of RFI ranged from 0.2 W m$^{-2}$ to 2.5 W m$^{-2}$, while under the aged snow assumption,
the estimated increase of RFI ranged from 0.9 W m$^{-2}$ to 5.7 W m$^{-2}$. During the Kuwait fires period, the
RFI values increased about 2-5 times higher than the "normal period", suggesting a significant increase
for the snow melting in Northern Tibetan Plateau due to this fire event. This result suggests that the
variability of BC deposition at the Muztagh Ata Mountain provides useful information to study the
effect of the upward BC emissions on environmental and climate issues in the Northern Tibetan Plateau.
The radiative effect of BC deposition on the snow melting provides important information regarding
the water resources in the region.

**Key Words; Northern Tibetan glaciers, BC deposition, MOZART model**



## 1 Introduction

Black carbon (BC) particles emitted from combustion are considered as an important air pollutant, as they change the radiative balance of the atmosphere directly by absorbing and scattering solar radiation, and indirectly by changing the microphysical process of cloud (acting as ice nuclei) and precipitation efficiency (acting as cloud condensation nuclei) (Ramanathan et al., 2001). Albedo changes induced by strongly light absorbing component by deposited on the surface of snow and ice are the key parameters to determine radiative forcing and accelerate melting (Holben et al., 1998; Hansen and Nazarenko, 2004). Due to the strong regional to local distribution of BC, these properties are not well understood, particularly in remote regions, such as the Tibetan Plateau.

BC particles can deposit and preserve in the ice by the progress of post-deposition on the glaciers and ice sheets. Retrieved ice cores from remote mountain glaciers and ice sheets provide useful information of the historical BC aerosol emissions and synchronous meteorology conditions. Previous studies on records of carbonaceous aerosols show that the emissions of fossil fuel combustion from Central Europe had significant impact on the glacier in the Swiss Alps (Lavanchy et al., 1999). Ice cores drilled from Antarctica suggest that the Southern Hemisphere biomass burning were strongly influenced by continental hydrology (Bisiaux et al., 2012). McConnell et al. (2007) differentiated the BC emissions from industrial activities and forest fires using an ice core in Greenland. These researches indicate that BC records in history are important and practicable method to investigate the regional aerosol transport and emission variations.

In this study, the ice core BC at Muztagh Ata, Northern Tibetan Plateau is analyzed. Identification the source regions, which have important impact on BC deposition at Muztagh Ata is very important scientific issue, because of its location. In particularly, there was a strong spike of the BC deposition during 1992-1993 at Muztagh Ata (as shown in the following text), reflecting that there was unusual emission in the upward region from Muztagh Ata. This strong spike of the ice core BC was about 3-4 times higher than other years, producing important effects on climate and hydrological cycle. As a result, the study of the sources of BC, which affect the ice core BC in this location, needs to be carefully studied. Muztagh Ata locates in the east of Pamir and the north of Tibetan Plateau. The ice core data provides important information for atmospheric circulation and climate change in Asia (An et al., 2001). Moreover, the climate in Muztagh Ata is very sensitive to solar warming mechanisms because it has a large snow cover in the region, resulting in important impacts on the hydrological cycle of the continent by enhancing glacier melt.

The BC sources which contribute the BC deposition in Tibetan Plateau have been previously studied. Their results show that BC deposited on glaciers in the Pamir Mountains was originated from Europe, Middle East and central Asia (Liu et al., 2008; Xu et al., 2009a; Wang et al., 2015b), whereas BC deposited on glaciers over the Himalayas and southeastern Tibetan Plateau was mainly affected by the western upward regions in winter. During the Indian summer monsoon season, they were mainly affected by the BC sources in Indian region (Ming et al., 2008; Xu et al., 2009b; Kaspari et al., 2011;





Wang et al., 2015a). However, at present, the effects of the transport pathways and individual
contributions of BC sources to the Muztagh Ata region have not been carefully studied. Because the
radiative forcing caused by BC in snow and ice between different regions is very different, depending
upon the emitting intensities, ocean-land distributions, topography, regional atmospheric circulations,
and other factors, detailed study on the source contributions to the region as well as the climate effect
are needed to carefully study this important region.

Both the ice core deposition measurements at Muztagh Ata and a global chemical model (MOZART-4;
Model for Ozone and Related chemical Tracers, version 4) are used in this study. To better evaluate the
model performance, the air samples of BC particles during 1986-1994 were also analyzed. The global
chemical transport model (MOZART-4) was used to analyze the long-term trend in the early 90s of the
observed BC deposition and to quantify the individual contribution of different BC sources to the
deposition on the snow cover. The modeled temporal variations and magnitude of the BC
concentrations in the atmosphere and snow were compared to observations. Finally, a radiative transfer
model (SNICAR) was used to study the effect of BC on snow albedo, radiative forcing, and runoff
changes induced by the BC deposition on the Muztagh Ata snow.

**2 Methodologies**
**2.1 Sampling Sites**
Muztagh Ata Mountain is located in the north side of Tibetan Plateau. Both atmospheric sampling and
ice core drilling BC were conducted at the Muztagh Ata site. The atmospheric sampling BC
(38°17.30'N, 75°01.38'E) was conducted by the Cold and Arid Regions Environmental and
Engineering Institute, Chinese Academy of Sciences, at a 4500 m above sea level (a.s.l.). A 170.4 m ice
core (9.5 cm in diameter) was drilled during the summer season in 2012 from Kuokuosele (KKSL)
Glacier of Muztagh Ata (38°11'N, 75°11'E, 5700 m a.s.l.), which was conducted by the Institute of
Tibetan Plateau Research, Chinese Academy of Sciences. Because the site is surrounded by several
important BC source regions, this measurement site is suitable to investigate the effect of BC emissions
on north part Tibetan Plateau, which plays important roles for global climate and hydrology (see Fig.

117  1).


The average annual temperature at the peak of the mountain is approximately -20$^{\circ}$C. Because the
numerous high mountains block the warm and humid air currents from Indian and Pacific Ocean, the
climate in this area is relatively dry. The averaged annual precipitation is less than 200 mm, which is
mainly snow to form perennial glaciers. There are 128 modern glaciers and on average about 377
square kilometers. The prevailing winds in this region are usually westerly jet stream. Previous studies
suggested that there was very small effect by local sources, and the aerosol pollutions were originated
mainly from the west by mid- and long-range transport. During summer, the South Asia monsoon had
also important effect on the transport of BC particles from India (Liu et al., 2008; Wu et al., 2008; Zhao



et al., 2011; Wang et al., 2015b).

**2.2 Measurements**

During the period from December 5, 2003 to February 17, 2006, Eighty-one valid total suspended
aerosol particle (TSP) and BC samples were obtained. The measurements were conducted under very
difficult environmental conditions, because of its high mountain location. The sample numbers for
spring, summer, autumn, and winter was 19, 21, 14, 27, respectively.   Each sample was collected over
one week and on 15 mm Whatman quartz microfibre filter (QM/A, Whatman LTD, Maidstone, UK),
which was pre-combusted at $800^{o}$C for 3 hours to remove the potential carbon disturbance.

For the ice core measurement, a 170.4 m ice core (9.5 cm in diameter) was drilled during the summer
season in 2012 from Kuokuosele (KKSL) Glacier of Muztagh Ata (38°11'N, 75°11'E, 5700 m a.s.l.),
which is close to the BC air sampling site. A 3-mm outer layer of the ice sections was removed with a
pre-cleaned stainless steel scalpel at -5$^{o}$C in a class 100 laminar flow bench to eliminate contamination
that may have occurred during drilling, transport, and storage. The inner section for BC analysis was
sealed in a 50 ml polypropylene vial (BD Falcon, cat. no. 358206). The ice core dating and calculation
of BC deposition fluxes were provided by Institute of Tibetan Plateau Research, Chinese Academy of
Science. The detailed method for the measurement of BC deposition is shown by Xu et al. (2009a).

**2.3 Measurements and analytical methods**

The elemental carbon (EC, which is proxy to BC in this study) analyses for atmospheric filters were
carried out by using Desert Research Institute (DRI) Model 2001 carbon analyzer (Atmoslytic Inc.,
Calabasas, CA, USA) with IMPROVE (Interagency Monitoring of PROtected Visual Environments)
thermal/optical reflectance (TOR) protocol (Chow et al., 1993; Chow et al., 2004). A 0.526 cm$^{2}$ punch
of a quartz filter sample was heated in a stepwise manner to obtain data for three elemental carbon (EC)
fractions (EC1, EC2, and EC3 in a 2% oxygen/98% helium atmosphere at 580, 740, and 840 $^{o}$C. At the
same time, OP (pyrolyzed carbon) was produced at <580 $^{o}$C in the inert atmosphere which decreases
the reflected light to correct for charred OC. Total EC is the sum of the three EC fractions minus OP.
More details and QAQC (Quality Assurance and Quality Control) are shown by Cao et al. (2003) and
Cao et al., (2009).

The rBC (refractory black carbon), which is used instead of BC for measurements derived from
incandescence methods (Petzold et al., 2013), was analyzed at Institute of Tibetan Plateau Research,
Chinese Academy of Sciences by using a Single Particle Soot Photometer (SP2) coupled with an
ultrasonic nebulization system (CETAC UT5000). The laser-induced incandescence was used to
measure the mass of rBC in individual particles (Schwarz et al., 2006). The incandescence signal can
be converted to rBC mass which is detected by photomultiplier tube detectors. This analytical method
was previously applied to ice cores by several studies ((McConnell et al., 2007; Kaspari et al., 2011;



Bisiaux et al., 2012). Detailed description on the SP2 analytical process and calibration procedures can
be found in (Wendl et al., 2014) and (Wang et al., 2015b).
Although the differences in the two analytical techniques (Wang et al., 2015b), in order to facilitate the
discussions, they are uniformly referred to as black carbon (BC) in our study since both of them are
materials share some of the characteristics of BC with its light-absorbing properties (Petzold et al.,
173  2013).

## 2.4 Global chemistry transport model / MOZART-4

The model used in this study is MOZART-4 (Model for Ozone and Related chemical Tracers, version
4). The model is an offline global chemical transport model for the troposphere developed jointly by
the National Center for Atmospheric Research (NCAR), the Geophysical Fluid Dynamics Laboratory
(GFDL), and the Max Planck Institute for Meteorology (MPI-Met). The detailed model description and
model evaluated can be found in Emmons et al. (2010). The aerosol modules was developed by Tie et
al. (2005). This model have been developed and used to quantify the global budget of trace gases and
aerosol particles, and to study their atmospheric transport, chemical transformations and removal
(Emmons et al., 2010; Chang et al., 2016).   The model is built base on the framework of the Model of
Atmospheric Transport and Chemistry (MATCH) (Rasch et al., 1997). Convective mass fluxes are
diagnosed by using the shallow and mid-level convective transport formulation of Hack (Hack, 1994)
and deep convection scheme (Zhang and McFarlane, 1995). Vertical diffusion within the boundary
layer is built on the parameterization by Holtslag and Boville (1993). Advective transport scheme used
the flux form semi-Lagrangian transport algorithm (Lin and Rood, 1996). The wet deposition includes
in-cloud as well as below-cloud scavenging developed by Brasseur et al. (1998) is taken into
MOZART-4. Details of the chemical solver scheme can be found in the Auxiliary Material (Kinnison
et al., 2007).
In the present study, the model includes 85 gas-phase species, 12 bulk aerosol compounds and
approximately 200 reactions. The horizontal resolution of this study is $1.9° \times 2.5°$ with 56 hybrid
sigma-pressure vertical levels from the surface to approximately 2 hPa. The meteorological initial and
boundary conditions are down load from NCAR Community Data Portal (CDP), using National
Centers for Environmental Prediction (NCEP) meteorology. The model transport of this study is driven
by the Modern-Era Retrospective-analysis for Research and Applications (MERRA) 6-hour reanalysis
data with a $1.9° \times 2.5°$ grid provided by National Aeronautics and Space Administration (NASA).
The BC emission inventory used in this global model is based on the simulation of     the POET
(Precursors of Ozone and their Effects in the Troposphere) database from 1997 to 2007 and the data of
BC emission inventory including fossil fuel and biofuel combustion from a previous study (Bond et al.,
2004; Bond et al., 2007). Figure 2 illustrates the updated 21-year average global BC emissions from
1986 to 2006. There are two types of black carbon particles in MOZART-4, hydrophobic and





hydrophilic particles. Hydrophobic particles are directly emitted from the sources, and are converted to
hydrophilic in the atmosphere (Hagen et al., 1992; Liousse et al., 1993; Parungo et al., 1994), with a
rate constant of $7.1 \times 10^{-6}$/s (Cooke and Wilson, 1996).


**2.5 BC deposition estimation**

In order to compare to the measured ice core BC deposition at the Muztagh Ata Mountain, the BC
deposition flux is calculated in this study. In the estimation, the calculated atmospheric BC
concentrations and precipitation data obtained from China Meteorological Data Service Center were
compiled and evaluated. In addition, annual BC deposition parameters and deposition flux calculation
methods were described in other studies (Jurado et al., 2008; Yasunari et al., 2010; Fang et al., 2015; Li
et al., 2017). In brief, deposition fluxes are calculated by the following equations:

$$F_{DD} = 10^{-4} v_D C_{BC}\, t \tag{1}$$
$$F_{WD} = 10^{-7} p_0 W_p C_{BC} \tag{2}$$
$$F_{BC} = F_{DD} + F_{WD} \tag{3}$$

where $10^{-4}$ and $10^{-7}$ are unit conversion factors; $F_{DD}$ and $F_{WD}$ are the annual dry and wet deposition
(ng cm$^{-2}$), respectively; the total BC deposition flux ($F_{BC}$) (ng cm$^{-2}$) is the sum of $F_{DD}$ and $F_{WD}$; where
$v_D$ (m s$^{-1}$) is the dry deoposition velocity of black carbon; $t$ is total estimation time for one year (s);
$p_0$ is the annual precipitation rate (mm); $W_p$ is the particle washout ratio (dimensionless); and $C_{BC}$ is
the annual atmospheric BC concentrations at Muztagh Ata Mountain (ng m$^{-3}$). There are large
differences in estimates on $v_D$ and $W_p$ (Jurado et al., 2005; Jurado et al., 2008; Yasunari et al., 2013).
A fixed small dry deposition velocity of $1.0 \times 10^{-4}$ m s$^{-1}$ onto snow was adopted (Yasunari et al., 2010;
Nair et al., 2013) and the corresponding estimation values are likely to represent a lower bound for BC
dry deposition in this area. Particle washout ratio $W_p$ is assumed to be a constant and equal to $2 \times 10^{5}$
which has been adopted in many modeling exercises and fits well with field measurements (Mackay et
al., 1986; Jurado et al., 2005; Fang et al., 2015; Li et al., 2017).
**3 Results and discussion**
**3.1 Model evaluation and compared to observation**

In order to better understand the variation, characteristics, and source contributions of the BC
concentrations at Muztagh Ata Mountain, model sensitive studies using MOZART-4 were conducted in
this study. Firstly, the model was evaluated by comparing the observed monthly BC concentrations
with the calculated monthly BC concentrations during January 2004 to February 2006. As shown in Fig
3a, the simulated BC concentrations had a similar magnitude of measured BC concentrations, with
mean values of 62.4 ng m$^{-3}$ and 56.5 ng m$^{-3}$ for the calculation and measurement, respectively. There




was also evident that the measured variability of BC was captured by the calculation. For example, the
calculated variability was comparable to the measured result between July 2014 and Oct. 2015.
However, some differences were also noticeable. For example, the calculated BC concentration was
overestimated in the winter of 2004 and underestimated in the winter of 2006. Because the measured
site locates in a "clean" region of BC emission, the BC particles were mostly transported from
long-distance of the upwind regions. There were uncertainty related to the emissions and simulated
meteorological parameters (wind speeds, wind directions, etc.). As a result, it caused the discrepancy
between calculated and measured BC concentrations at the Muztagh Ata Mountain. There was another
reason may cause the difficulty of the calculation. The horizontal resolution of the global model is
relatively low ($1.9° \times 2.5°$ in this study), which is unable to reproduce some detailed variability in the
simulation. However, the overall features of the measured BC concentrations were reproduced by the
model, such as the magnitude and seasonal variability (see Fig. 3b), suggesting that the model is
capable to study long-range transport from BC source regions to the remote site.

The simulated seasonal variation shows in Fig 3b. The result shows that calculated seasonal variation
was generally agreed with the measured variation, except the value in spring. According to the analysis
of the source contribution (shown in Section 3.3), the BC emission in South Asia has significant
contributions to the BC concentrations at Muztagh Ata during non-summer season which accounted for
average 31~60% in spring and few contributions in summer season. The overestimated BC
concentrations may due to the fact that the model overestimated the pollutant transportation from the
emission sources to sampling site crossing the high mountains of Tibet Plateau, which act as a wall to
block the transportation from the BC emission in South Asia to the sampling area (Zhao et al., 2013).

**3.2 Long-term Ice core measurement and possible effect of Kuwait fire event**

In addition to the atmospheric sampling of BC measurement, there is a long-term ice cores
measurement of BC at the Muztagh Ata Mountain. This long-term measurement represents a valuable
data to show the long-term trend and inter-annual variability. Ice core records obtained at Muztagh Ata
Mountain are irreplaceable when evaluating contemporary atmospheric or snow BC concentration
variations. A long-term ice-core measurement (from 1940 to 2010) was provided by Xu et al. at
Muztagh Ata Mountain. Their results showed that the ice core BC concentrations were between 0.30
and 39.54 ng $g^{-1}$ from 1940 to 2010, with an average value of 7.22 ng $g^{-1}$. The BC deposition fluxes
were between 9.96 and 909.88 ng $cm^{-2}$, with an average of 184.18 ng $cm^{-2}$. It is interesting to note that
both BC concentration and BC deposition of ice core showed a sharply increase in 1992, which was
about five times higher than the average mean value as shown in Fig 4. No other similar peak was
found in the entire record which may indicate a specific event to lead to this sharp increased, which
provide useful information to track the BC emissions. In this study, we conduct several model studies
to investigate this special event.

As shown in Figure 4, there was a high BC deposition flux (900 ng $cm^{-2}$) in 1992, compared to 100-300



ng cm⁻² in other years. In order to investigate this special event, we focus our model study on a short
period (from 1986 to 1994). One potential reason to cause this sharp increase of BC was that during
1991, when Iraqi troops withdrew from Kuwait at the end of the first Gulf War, they setted a huge fire
over 700 oil wells. The fires were started in January and February 1991, and the last well was capped
on November 6, 1991. The resulting fires produced a large plume of smoke and particles that had
significant effects on the Persian Gulf area and the potential for global effects (as shown in Fig. 5).

In order to estimate intensive of the BC emission from the fires, (Hobbs and Radke, 1992) conducted
two aircraft studies during the period 16 May through 12 June 1991 to evaluate the effects of the smoke.
The estimated emission rate of elemental carbon of the Kuwait fires is ~3400 metric tons per day which
is 13 times the BC emissions from all U.S. combustion sources in total.

In order to the effect of the huge Kuwait fires on the BC ice core deposition, the MOZART-4 model
was applied to simulate the atmospheric BC concentrations and deposition fluxes variation from 1986
to 1994. Several model sensitive studies were conducted. First, the atmospheric BC concentration was
calculated by the anthropogenic BC emission with the default emissions (POET) as described before.
Second, in order to simulate the large increase in the BC emissions caused by the Kuwait fires in
Persian Gulf (Region 3 in Fig. 1), according to the measured values of Hobbs and Radke (1992), the
BC emissions were significantly enhanced by 50 times from January to November, 1991 to represent
Kuwait fires. Figure 4 shows the horizontal distribution of the calculated BC plume from the Kuwait
fires, with the enhanced BC emission.


The calculated result suggests that there was a significant increase of BC concentrations nearby the
Kuwait fires (see Fig. 6). The BC concentrations reached to 10-20 μg m⁻³ at the surface (see Fig. 6A)
and more than 0.7 μg m⁻³ at 5 km above the surface (see Fig. 6B). As shown in Figs. 5 and 6, the winds
nearby the fire region were toward to northern and northwestern directions. Because the lifetime of
black carbon aerosols is sufficiently long (about a week) (Ramanathan et al., 2001; Bauer et al., 2013),
the high BC concentrations were transported westerly toward the Muztagh Ata Mountain.

The evaluation of the modeled BC deposition at the Muztagh Ata Mountain was conducted by
comparison between the calculation and measurement (see Fig. 4). Figure 4 shows the calculated
temporal variation of BC concentrations and deposition, which were compared with the measured
variations. The result shows that the calculated temporal variability of BC deposition was generally
consistent with the measured variability. For example, the both high peaks of calculated and measured
BC deposition occurred in 1992. The calculated atmospheric concentrations of BC, however, had a
peak value in 1991. This was due to the fact that the deposition of BC in ice core was an accumulated
value, while the atmospheric BC concentration was an in-situ value. Despite of the consistence of
temporal variations between measured and calculated deposition of BC, there was a consistent
underestimate of calculated BC deposition compared to the measured value. Because there were





uncertainties in estimates BC emission and the deposition, these uncertainties could result in the
discrepancy between the calculation and measurement. For example, according to the assimilation
meteorological data by Chinese Meteorological Admiration, the annual precipitation in 1992 was about
twice higher than in 1991 nearby Muztagh Ata Mountain, suggesting that scavenging efficiency may
likely underestimated, causing the calculated uncertainty in the estimate of the BC deposition.

**3.3  Effect of regional BC emissions at the Muztagh Ata Mountain**

To further understand the influence of transportation and deposition on the annual variation of BC at
the Muztagh Ata Mountain (as a receptor region), sensitivity experiments using the MOZART-4 model
were conducted. In the sensitive study, the effect of different BC emission regions on the BC
concentrations at the measurement site was individually calculated. Four primary regions were defined
as shown in Table 1 and Fig. 1, including (R1) Central Asia, (R2) Europe, (R3) Persian Gulf, and (R4)
South Asia. In each sensitive study, only the individual BC emission was included, and the BC
emissions in other regions were excluded. As a result, the fractional contributions by the individual
emission regions to BC concentrations in the receptor region (the Muztagh Ata Mountain) were
calculated. Table 1 shows the calculated results.

In order to clearly show the transport pathways from the different regions to the measurement site and
the Tibetan Plateau, the calculated horizontal distributions of BC concentrations from each region
during 3 different periods (summer monsoon, non-monsoon, and annual mean) were shown in Fig. 7.

The results from Table 1 and Fig. 7 suggests that during the "normal period" (non Kuwait Fires), the
BC emissions from Central Asia and South Asia had the largest contributions to the BC concentrations
at measurement site, contributing annual mean of 27% and 25%, respectively. It is interesting to note
that there were strong seasonal variations regarding the effects. During the monsoon period, the largest
effect was due to the Central Asia source (44%), while during non-monsoon period, the largest effect
was due to the South Asia source (34%).

As shown in Fig. 7, during the monsoon period, the airflow from the oceans (Persian Gulf and Bengal
Bay) moves northward and coupled with the strong precipitation.   As a result, the BC particles from
south Asian source were washout during the transport pathway, leading to lower BC concentrations at
the measurement site. In contrast, during the non-monsoon period, the prevailing winds were western
winds, which BC emission in the northern India was transported to the measurement site measurement
site, leading to higher BC concentrations. The contributions from Persian Gulf emissions were
generally low to the BC concentrations. However during Kuwait fires period, this region had
significant contribution to the Muztagh Ata area as well as the Tibetan Plateau.

**3.4  Radiative forcing induced by BC in Muztagh Ata glacier**





The deposition of BC on the snow reduces the surface albedo, causing a positive radiative forcing and
increases in ice and snow melt. Previous studies show that BC particles produce significant reduction in
the snow albedo, with the solar visible wavelengths (Warren and Wiscombe, 1980). In this study, the
effect of BC deposition on the snow albedo and radiative forcing during 1986 to 1994 in Muztagh Ata
glacier was estimated. The SINICAR model (Snow, Ice, and Aerosol Radiation; available at
http://snow.engin.umich.edu) was used to estimate the effect of BC particles on snow albedo in
different solar wavelengths (Flanner and Zender, 2005; Flanner et al., 2007).

To estimate the effect of the BC deposition on surface albedo, in addition to the BC concentrations,
there are several environmental factors such as snow grain size, solar zenith angle, and snow depth
were needed to be estimated (Warren and Wiscombe, 1980). The setup of input parameters required for
running the SNICAR model is briefly described as below. As we focus on the calculation of radiative
forcing caused by BC particles, other impurity contents, such as dust and volcanic ash, were set to be
zero. A mass absorption cross section (MAC) of 7.5 $m^2$ $g^{-1}$ at 550 nm for uncoated BC particles (Bond
and Bergstrom, 2006) was assumed to be same as the default value, and the MAC scaling factor in the
online SNICAR model as one of input parameters was set to be 1.0. The effective radius of 100 μm
with a density of 60 kg $m^{-3}$ was used for new snow, and the effective radius of 400 μm with a density of
400 kg $m^{-3}$ was adopted for the albedo estimation according to the previous studies and measurements
in other studies in Tibetan Plateau (Wiscombe and Warren, 1980; Wu et al., 2006). The extractive snow
height from MERRA (the Modern-Era Retrospective-analysis for Research and Applications)
reanalysis products was used for snowpack thickness. The forcing dataset used in this study was
developed by Data Assimilation and Modeling Center for Tibetan Multi-spheres, Institute of Tibetan
Plateau Research, Chinese Academy of Sciences (Chen et al., 2011). The recovered BC concentrations
of ice core were used as the input parameter of uncoated black carbon concentration. The averaged
short-wave flux and solar zenith angle of each month were obtained from China Meteorological
Forcing Dataset provided by Data Assimilation and Modeling Center for Tibetan Multi-spheres,
Institute of Tibetan Plateau Research, Chinese Academy of Sciences.

The measured average BC concentration in ice core during 1986-1994 was 15.2 ng $g^{-1}$, with a peak
value of 39.2 ng $g^{-1}$. The calculated snow albedo reduction by using the SNICAR model ranged from
0.11% to 1.36% by assuming that the snow layer was totally covered by fresh snow (lower limit).
However, if it was aged layer, the estimated snow albedo reduction increased, ranging from 0.47% to
2.97% (upper limit). The actual value should be lied between the two ranges. This result is consistent
with the previous studies. For example, (Yasunari et al., 2010) reported that the reduction of snow
albedo ranged from 2.0% to 5.2%, with the BC concentration of 26.0-68.2 ng/g, based on atmospheric
BC measurements at NCO-P over the southern slopes of western Himalayas.

The reduction of snow albedo enhanced the absorption of solar energy and accelerated snow and ice
melt (Conway et al., 1996). Several studies suggested that that BC containments on snow were very
effective to reduce the surface albedo (Warren and Wiscombe, 1980; Petr Chylek and Srivastava, 1983;





405 Gardner and Sharp, 2010). In this study, the effects of BC containments on snow albedo and snow

406 water equivalent (SWE) reduction were estimated.


408 Figure 8 shows the calculated the effects of BC containments on annual mean radiative forcing increase

409 (RFI) (W m$^{-2}$) and snow water equivalent (SWE) reduction (mm yr$^{-1}$; millimeter per year), under fresh

410 snow assumption and aged snow assumption. The results show that under the fresh snow assumption

411 (lower limit), the increases of RFI ranged from 0.2 W m$^{-2}$ to 2.5 W m$^{-2}$, while under the aged snow

412 assumption (upper limit), the increases of RFI ranged from 0.9 W m$^{-2}$ to 5.7 W m$^{-2}$. This estimate is

413 consistent with the previous studies (Flanner et al., 2009)During the Kuwait fires period, the RFI values

414 increased about 2-5 times higher, which led to a significant increase for the snow melting during the

415 period.

416

417 The runoff of the melted snow due to the increase of snow surface albedo was estimated in this study.

418 A first-order estimation was based on the additional energy contribution to the snowpack due to BC

419 deposition. First the melting point of snow was estimated. Second, the extra snow melt from light

420 absorbing black carbon was estimated by dividing hourly instantaneous radiative forcing, with the

421 enthalpy of fusion of water at 0 $^{\circ}$C of 0.334 $\times$ 10$^6$ J kg$^{-1}$ (Painter et al., 2013; Kaspari et al., 2015).

422 The estimation represented the snow melt in kg m$^{-2}$ across the hour during acquisition translates, or

423 melt in mm of snow water equivalent (SWE). The melted snow due to the BC water was calculated

424 (shown in Fig. 8). The result shows that the estimated averaged SWE reductions were 111 mm and 270

425 mm, for fresh and aged snow respectively. During the Kuwait fires period, the estimated SWE

426 significantly increased, reaching to 600 mm for aged snow condition, and 300 mm for fresh snow

427 condition. The increase was about 3 times than pre- and post- Kuwait fires, suggesting that this special

428 event had a significant impact on snow melting for the Tibetan glaciers and the water resources in the

429 region.

430 **4 Conclusions**

431 Black carbon (BC) particles change the radiative balance of the atmosphere by absorbing and scattering

432 solar radiation. As a result, BC deposition on the surface of snow and ice changes the albedo of solar

433 radiation. Albedo change is the key parameter to affect the melting of glacier in Tibetan Plateau. In

434 order to study this effect, two sets of measurements were used to study the variability of BC deposition

435 at Muztagh Ata Mountain, Northern Tibetan Plateau. The measured data included the air samplings of

436 BC particles during 2004-2006 and the ice core drillings of BC deposition during 1986-1994. To

437 identify the effect of BC emissions on the BC deposition in this region, a global chemical

438 transportation model (MOZART-4) was used to analyze the BC transport from the source regions. A

439 radiative transfer model (SNICAR) was used to study the effect of BC deposition on snow albedo.

440

441 The results show some important highlights to reveal the temporal variability of BC deposition and the

442 effect of long-rang transport on the BC pollution in the Northern Tibetan Plateau, which are





summarized as the follows;

(1) During 1991-1992, there was a strong spike of the BC deposition at Muztagh Ata, suggesting
that there was unusual emission in the upward region. This high peak of BC deposition was
investigated by using the global chemical transportation model (MOZART-4). The analysis
indicated that the emissions from large Kuwait fires at the end of the first Gulf War in 1991
caused the high peak of the BC concentrations and the BC deposition. As a result, the BC
deposition in 1991 and 1992 at the Muztagh Ata Mountain was 3-4 times higher than other
periods.

(2) The effect of Kuwait fires on the BC deposition at the Muztagh Ata Mountain suggested that
the upward BC emissions had important impacts on this remote site located in Northern
Tibetan Plateau. In order to quantitatively estimate the effect of surrounding emissions on the
BC concentrations in the northern Tibetan Plateau, a sensitive study with 4 individual BC
emission regions (Central Asia, Europe, Persian Gulf, and South Asia) was conducted by
using the MOZART-4 model. The result suggests that during the "normal period" (non Kuwait
Fires), the largest effect was due to the Central Asia source (44%) during Indian monsoon
period. During non-monsoon period, the largest effect was due to the South Asia source
(34%).

(3) The increase of radiative forcing increase (RFI) due to the deposition of BC on snow was
estimated by using the radiative transfer model (SNICAR). The results show that under the
fresh snow assumption, the estimated RFI ranged from 0.2 W m$^{-2}$ to 2.5 W m$^{-2}$, while under
the aged snow assumption, the estimated RFI ranged from 0.9 W m$^{-2}$ to 5.7 W m$^{-2}$. During the
Kuwait fires period, the RFI values increased about 2-5 times higher than the "normal period",
suggesting a significant increase for the snow melting in Northern Tibetan Plateau due to this
fire event.


This result suggests that the variability of BC deposition at the Muztagh Ata Mountain provides useful
information to study the effect of the upward BC emissions on environmental and climate issues in the
Northern Tibetan Plateau. The radiative effect of BC deposition on the snow melting provides
important information regarding the water resources in the region.

**Acknowledgement**
This work was supported by the National Natural Science Foundation of China (NSFC) under Grant
Nos. 41430424 and 41730108. The Authors thanks the supports of Center for Excellence in Urban
Atmospheric Environment, Institute of Urban Environment, Chinese Academy of Sciences. The
National Center for Atmospheric Research is sponsored by the National Science Foundation.






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



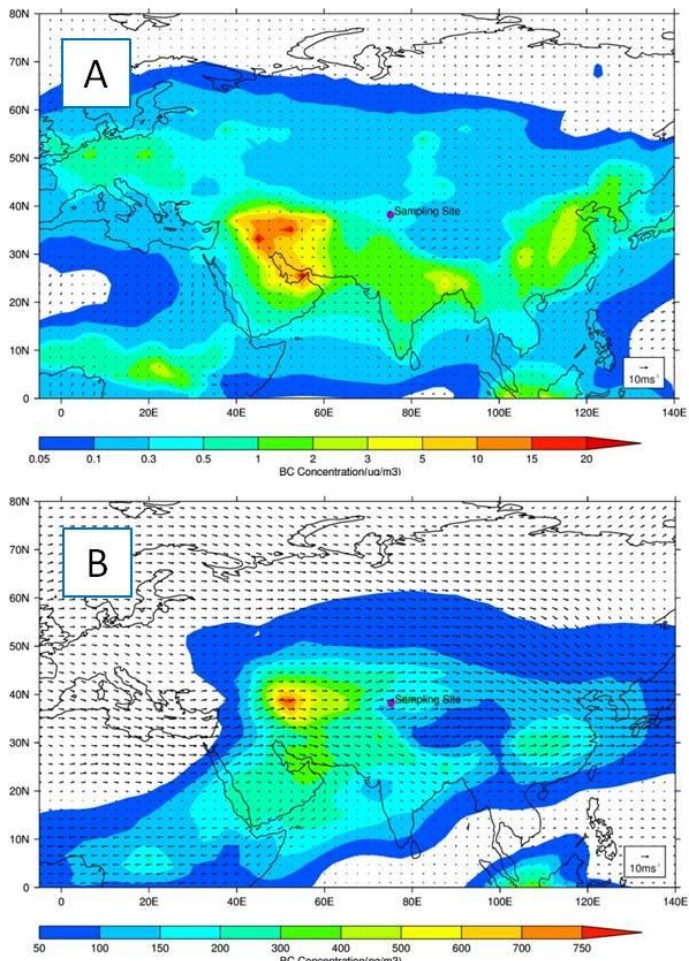

Fig. 6. The calculated horizontal distributions of BC concentrations ($\mu g\ m^{-3}$) at the surface (A) and the
concentrations ($ng\ m^{-3}$) at 5 km above the surface (B). The wind direction and speed are indicated by black
arrows.



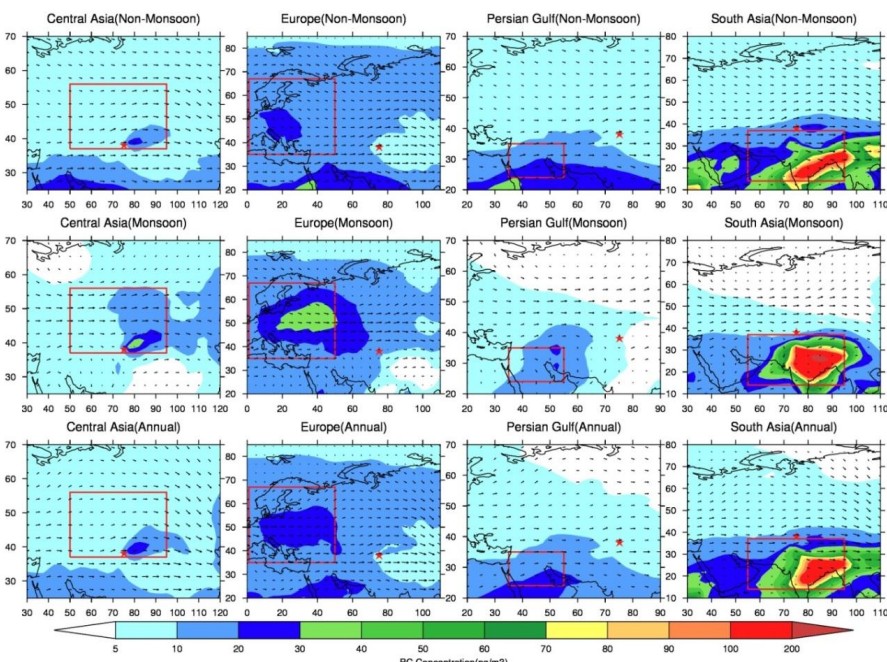

**Fig 7. The calculated spatial BC distributions due to individual BC from the four source regions (Central Asia, Europe, Persian Gulf and South Asia) above 5 km above the surface          during different periods, i.e., monsoon (June-September), non-monsoon (October-May), and annual mean in 1993. The red star is where the study site of Muztagh Ata located. The red boxes indicate the boundary of the four source regions.**





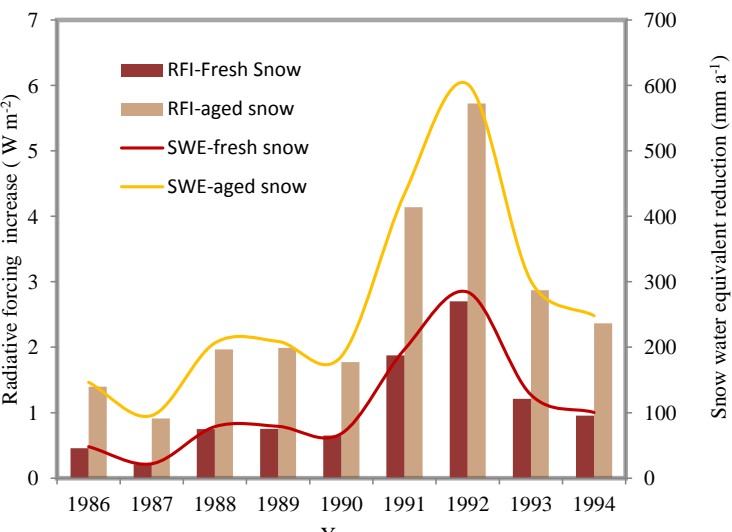


**Fig 8. Estimated the effects of BC containments on annual mean radiative forcing increase (RFI) (W/m²)**

**and snow water equivalent (SWE) reduction (mm/a), under fresh snow assumption (purple line and bars)**

**and aged snow assumption (yellow line and bars).**



**Table 1. Source regions and corresponding fractional contributions to atmospheric BC concentrations at the**

**Muztagh Ata site in monsoon, non-monsoon and all months during 1993**


|  | Source regions | Latitude | Longitude | Summer monsoon (June-September) | Non-monsoon (October-May) | Annual |
|---|---|---|---|---|---|---|
| R1 | Central Asia | 37-56 °N | 50-95 °E | 43.9% | 18.1% | 26.7% |
| R2 | Europe | 35-67 °N | 0-50 °E | 26.6% | 11.5% | 16.5% |
| R3 | Persian Gulf | 24-35 °N | 35-55 °E | 9.4% | 12.1% | 11.2% |
| R4 | South Asia | 14-37 °N | 55-95 °E | 7.3% | 33.7% | 24.9% |
|  | Others | NA | NA | 7.9% | 6.2% | 6.8% |




