# Peer review of "Black carbon (BC) in North Tibetan Mountain; Effect of Kuwait fires on glacier"

_Atmospheric Chemistry and Physics, 2018_

## Referee Comment (RC1) · Anonymous Referee #1 · 28 May 2018

The manuscript studies the BC deposition and its radiative effect on the snow cover in the northern Tibetan Plateau. Two sets of measurements were used in this study, which included the air samplings of BC particles during 2004-2006 and the ice core drillings of BC deposition during 1986-1994. These data are very interesting and valuable. In addition, two numerical models are used in this study to analyze the data, including; a global chemical transportation model (MOZART-4) and a radiative transfer model (SNICAR). Their analysis shows that there is a high peak of BC deposition at Muztagh Ata in Northern Tibetan Plateau during 1991-1992 (about 3-4 times higher than other years), caused by the large Kuwait fires at the end of the first Gulf War in 1991. This result suggests that the upward BC emissions had important impacts on this remote site located in Northern Tibetan Plateau. The radiative effect calculated by the radiative

transfer model (SNICAR) shows that a significant increase for the snow melting in Northern Tibetan Plateau due to this fire event. This study is suitable for the scientific scope of ACP, and can be accepted for the publication in ACP. However, there are some minor comments, which should be addressed in the revised version;

Comments; (1) The Authors define 4 BC source regions, which could have important impacts on the BC deposition at the remote site located in Northern Tibetan Plateau. They should make more detailed description for the definition of these 4 regions. (2) The Authors have detailed description for the ice core drill measurements. However, the description of TSP is rather too simple. More descriptions of the TSP should be required. (3) The quality of Fig. 6 should be improved. The labels are too small. (4) There are some English typos. For example, in the line 297, Page 9, "In order to the effect of the huge Kuwait fires on the BC ice core deposition" should be "In order to study the effect of the huge Kuwait fires on the BC ice core deposition"

---

## Referee Comment (RC2) · Anonymous Referee #2 · 10 Aug 2018

This articel investigate the large Kuwait fires on BC deposition on the ice core at Muztagh Ata Mountain, Northern Tibetan Plateau and the related radiative forecing. It has excellent scientific point and is meaningful for the current Tibetan Plateau experiments. I strongly suggest the acceptance and qulick publishment of the articel. Following is some comments and suggestions for the paper:

(1) In Fig.1, the topography should be plotted to illustrate the plateau characteristics.

(2) In Fig.2, the BC measurements were much lower during Apr to May of 2004, and sharply increased on Jun, while the model results were very flat, the author should give some explanations.

(3) I suggest the author made more discussion on the possible impact of the change of

ice on regional climate, such as the flood, the drought in china.

---

## Author Comment (AC1) · 23 Aug 2018

Reviewer 1:

We thank the reviewer for the careful reading of the manuscript and helpful comments. We have revised the manuscript following his/her suggestions as is described below.

Reviewer #1: The manuscript studies the BC deposition and its radiative effect on the snow cover in the northern Tibetan Plateau. Two sets of measurements were used in this study, which included the air samplings of BC particles during 2004-2006 and the ice core drillings of BC deposition during 1986-1994. These data are very interesting and valuable. In addition, two numerical models are used in this study to analyze the data, including; a global chemical transportation model (MOZART-4) and

a radiative transfer model (SNICAR). Their analysis shows that there is a high peak of BC deposition at Muztagh Ata in Northern Tibetan Plateau during 1991-1992 (about 3-4 times higher than other years), caused by the large Kuwait fires at the end of the first Gulf War in 1991. This result suggests that the upward BC emissions had important impacts on this remote site located in Northern Tibetan Plateau. The radiative effect calculated by the radiative Only one month sampling of PM2.5 was conducted in this study, which cannot view the current status of atmospheric fine PM2.5. At least four seasons are commonly required in a typical PM2.5 study. transfer model (SNICAR) shows that a significant increase for the snow melting in Northern Tibetan Plateau due to this fire event. This study is suitable for the scientific scope of ACP, and can be accepted for the publication in ACP. However, there are some minor comments, which should be addressed in the revised version;

Comments; (1) The Authors define 4 BC source regions, which could have important impacts on the BC deposition at the remote site located in Northern Tibetan Plateau. They should make more detailed description for the definition of these 4 regions.

Response: To address the reviewer's comments, we define the 4 sources regions with a detailed description. The corresponding revision can be found from the line 414 to 422. We also plotted the topography of the study region as shown in Fig.1.

(2) The Authors have detailed description for the ice core drill measurements. However, the description of TSP is rather too simple. More descriptions of the TSP should be required.

Response: According to the suggestion, we added the information of the samplers of TSP, including sampling flow rates, power of device and the identification of valid samples from the line 163 to 172. The description of ice core drill measurements in section 2.2 has been revised correspondingly.

(3) The quality of Fig. 6 should be improved. The labels are too small.

Response: Fig.6 has been improved as request.

(4) There are some English typos. For example, in the line 297, Page 9, "In order to the effect of the huge Kuwait fires on the BC ice core deposition" should be "In order to study the effect of the huge Kuwait fires on the BC ice core deposition"

Response: Corrected. We've also checked other typos and make corrections in the revised version.

---

## Author Comment (AC2) · 23 Aug 2018

Reviewer 2:

We thank the reviewer for the careful reading of the manuscript and helpful comments. We have revised the manuscript following his/her suggestions as is described below.

Reviewer #2: This articel investigate the large Kuwait fires on BC deposition on the ice core at Muztagh Ata Mountain, Northern Tibetan Plateau and the related radiative forecing. It has excellent scientific point and is meaningful for the current Tibetan Plateau experiments. I strongly suggest the acceptance and qulick publishment of the articel. Following is some comments and suggestions for the paper:

(1) In Fig.1, the topography should be plotted to illustrate the plateau characteristics.

[Figure]

Response: The topography of Fig.1 has been updated.

(2) In Fig.2, the BC measurements were much lower during Apr to May of 2004, and sharply increased on Jun, while the model results were very flat, the author should give some explanations.

Response: To address the reviewer's comment, we make explanation that the difference between the measured and the modeled BC concentrations during the spring of 2004 is due to the ucertainties of the emissions, simulated meteorological parameter and the low horizontal resolution, which lead to difference of topography between the model and actual situation. These explanations can be added from line 309 to 320.

(3) I suggest the author made more discussion on the possible impact of the change of ice on regional climate, such as the flood, the drought in china.

Response: Thanks for the constructive suggestion from reviewer. We've added discussion from the line 516-542.